# INO80 Is Required for the Cell Cycle Control, Survival, and Differentiation of Mouse ESCs by Transcriptional Regulation

**DOI:** 10.3390/ijms232315402

**Published:** 2022-12-06

**Authors:** Seonho Yoo, Eun Joo Lee, Nguyen Xuan Thang, Hyeonwoo La, Hyeonji Lee, Chanhyeok Park, Dong Wook Han, Sang Jun Uhm, Hyuk Song, Jeong Tae Do, Youngsok Choi, Kwonho Hong

**Affiliations:** 1Department of Stem Cell and Regenerative Biotechnology, Institute of Advanced Regenerative Science, Konkuk University, Seoul 05029, Republic of Korea; 2Guangdong Provincial Key Laboratory of Large Animal Models for Biomedicine, Wuyi University, Jiangmen 529020, China; 3Department of Animal Science, Sangji University, Wonju 26339, Republic of Korea

**Keywords:** ES cell, Ino80, cell cycle, apoptosis, differentiation

## Abstract

Precise regulation of the cell cycle of embryonic stem cells (ESCs) is critical for their self-maintenance and differentiation. The cell cycle of ESCs differs from that of somatic cells and is different depending on the cell culture conditions. However, the cell cycle regulation in ESCs via epigenetic mechanisms remains unclear. Here, we showed that the ATP-dependent chromatin remodeler Ino80 regulates the cell cycle genes in ESCs under primed conditions. Ino80 loss led to a significantly extended length of the G1-phase in ESCs grown under primed culture conditions. Ino80 directly bound to the transcription start site and regulated the expression of cell cycle-related genes. Furthermore, Ino80 loss induced cell apoptosis. However, the regulatory mechanism of Ino80 in differentiating ESC cycle slightly differed; an extended S-phase was detected in differentiating inducible Ino80 knockout ESCs. RNA-seq analysis of differentiating ESCs revealed that the expression of genes associated with organ development cell cycle is persistently altered in Ino80 knockout cells, suggesting that cell cycle regulation by Ino80 is not limited to undifferentiated ESCs. Therefore, our study establishes the function of Ino80 in ESC cycle via transcriptional regulation, at least partly. Moreover, this Ino80 function may be universal to other cell types.

## 1. Introduction

Compared with somatic cell growth, embryonic stem cell (ESC) growth has unique features, including a very short Gap1 (G1) phase and a slightly prolonged DNA synthesis (S) phase [1]. This ESC-specific cell cycle is mediated by the differential expression or activity of cell cycle controllers. For example, unlike somatic cells, mouse ESCs (mESCs) exhibit a typical oscillatory activity of cyclin-dependent kinase (Cdk) complexes, low levels of cyclins, enhanced Cdk activity, and a lack of Cdk inhibitors (CKIs) [1,2,3]. Additionally, ESCs cultured in 2i condition exhibit a considerably longer G1-phase compared with those cultured in normal serum conditions [4,5]. During the G1-phase, mESCs express low levels of cyclins D1 and D3, and they display high activity of cyclin E/Cdk2 and cyclin A/Cdk2 throughout the ESC cycle [4,6,7,8]. Cdk2 is a factor that causes ESCs to enter S-phase rapidly, and its activity is inhibited by p21 and p27 [4]. When comparing the levels of cell cycle regulators between serum-cultured ESCs (serum–ESCs) and 2i-cultured ESCs (2i–ESCs), the levels of CDK2, CDK4, CDK6, CCND, and CCNE slightly differed in the two groups, whereas much higher levels of CDK inhibitors (P16, P21, and P27) were detected in 2i–ESCs than in serum–ESCs during the late G1-phase [4]. Given that the growth of ESCs maintained in 2i condition is much slower than that in serum condition, levels of the CKIs are considered a critical contributor to the length of the ESC cycle. In addition, the activity of retinoblastoma protein (pRB), a deterministic cell cycle regulator of the G1-phase in serum–ESCs, is regulated by P21 and P27 [4]. Furthermore, cell cycle-regulated activity of G1-phase cyclins is necessary for cell fate determination through the activation of developmental genes [8]. However, the precise mechanism behind the shortened G1-phase in ESCs by chromatin remodelers is not completely understood yet.

Ino80 is an ATP-dependent chromatin remodeler that plays indispensable roles in development, cancer progression, and gene transcription [9,10,11,12,13,14,15,16,17,18,19,20,21,22,23,24,25,26,27,28]. Germline deletion of Ino80 leads to embryonic lethality due to failure in gastrulation, growth retardation, and massive cell death [14,15,23]. The Ino80 complex is highly enriched at the transcription start site (TSS) and can either promote or repress gene expression in a context-dependent manner [17,24,25,27,28,29]. In the promotion of gene expression, Ino80 permits Mediator and RNA polymerase II (RNAPII) to be recruited for gene activation while maintaining an open chromatin architecture [24]. A component of the Ino80 complex, Yin-Yang-1, functions as a transcription factor and promotes gene transcription in mouse E13.5 cerebral cortex and HEK293 cells [27,29]. A recent study showed that Ino80 participates in the stability of the +1 nucleosome, helps RNAPII pausing site determination, and ensures correct early transcription elongation [25]. However, Ino80 also participates in transcriptional repression in mESCs and spermatocytes; it binds to the TSS and recruits the H3K27me3 repressive mark to contribute to transcriptional repression [17,28].

Ino80 function, mainly the regulation of gene transcription, is essential for the maintenance of pluripotent stem cells [24,28]. Yu et al. showed that Ino80 builds a bivalent chromatin structure in developmental genes of primed ESCs by modulating histone H2A.Z occupancy-mediated PRC2 complex recruitment to the genes. This leads to the repression of the developmental genes in primed ESCs and is dominant, as this Ino80 function is absent in naïve ESCs [28]. Moreover, previous studies have commonly argued that Ino80 loss leads to cell death and growth retardation in mESCs, but its underlying mechanism remains elusive [14,28].

In the present study, we showed that Ino80 directly regulates the cell cycle of undifferentiated and differentiating ESCs via transcriptional repression of genes such as Cdkn1a and Rb1. Its function is not limited to undifferentiated ESCs, since cell death and aberrant gene expression was also observed in differentiating ESCs. Therefore, our study provided some evidence that the ESC cycle is potentially regulated by epigenetic mechanism(s).

## 2. Results

### 2.1. Ino80 Loss in ESCs Causes Aberrant Cell Cycle and Cell Death

To understand the Ino80 function in ESCs, inducible Ino80 knockout (Ino80 iKO) ESCs were established from Ino80^2f/2f^;ROSA26-CreER blastocysts. ESCs established from Ino80^2f/2f^ blastocysts were used as control ESCs (CONT). Next, to investigate the role of Ino80 in ESC maintenance, 4-hydroxytamoxifen (4-OHTM) was administered for three days. Ino80 deletion was confirmed by PCR genotyping for the null allele and qRT-PCR for the Ino80 transcripts (Figure 1A,B). Immunofluorescence analysis revealed no difference in the expression of Nanog, Oct4, Sox2, or SSEA1 between CONT and Ino80 iKO ESCs without 4-OHTM treatment (Figure 1C). This indicated that the maintenance of the established Ino80 iKO ESCs is not altered without 4-OHTM treatment.

Consistent with previous reports, Ino80 iKO ESCs maintained in either 2i- or serum-media were less proliferative and underwent cell death following 4-OHTM treatment [17,22,23,26,27,28]. However, despite the comparable expression levels of Ino80 in both culture conditions, the phenotypes in 2i–ESCs were less severe than those in serum–ESCs after Ino80 deletion (Figure 1B,D; microscope analysis). To measure cell growth in more detail, 20,000 4-OHTM-treated ESCs were seeded and counted at days 2, 3, and 4 after seeding. Significant retardation of cell growth was detected in serum–iKO ESCs (Figure 1E). This finding was further confirmed by flow cytometric analysis. Flow cytometry revealed significant cell death in serum–iKO ESCs (Figure 1F). Interestingly, a more extended G1-phase was also detected in the Ino80 iKO ESCs than in CONT ESCs, and the extension of the G1-phase depended on cell culture conditions; the extension of the G1-phase was much greater in serum–ESCs than in 2i–ESCs (8.47% in 2i–ESCs vs. 12.96% in serum–ESCs) (Figure 1G, Appendix A). Panels of flow cytometric analysis of cell cycle were included in Appendix A. The serum–iKO ESCs included considerably more apoptotic cells than 2i–ESCs. The differential expression of Cdkn1a, a cell cycle arrest-related gene, was confirmed by qRT-PCR (Figure 1H).

### 2.2. Ino80 Loss Alters Expression of Genes Related to Cell Cycle

We analyzed the underlying mechanism by which Ino80 loss causes cell cycle arrest and apoptosis. First, publicly available RNA-seq (GSE158545) and ChIP-seq (GSE158454) datasets were obtained and reanalyzed. Our analysis identified 5483 differentially expressed genes [(DEGs; cutoff values: fold change (FC) ≥2, fragments per kilobase of transcript per million (FPKM) ≥2)] with four different groups (Figure 2A). Among these groups, groups 2 and 4 showed unique patterns of gene expression profiles. Group 2 (801 genes) was more repressed in primed ESCs by Ino80 deletion, including genes related to amino acid biosynthesis, translation, and cell cycle, whereas group 4 (622 genes) was enhanced in primed ESCs by Ino80 deletion, including genes related to cell cycle arrest, apoptosis, and transcription. Then, we examined whether the genes associated with cell cycle and apoptosis were direct targets of Ino80. The analysis revealed that Ino80 bound to the TSS of 30 out of 44 cell cycle-related genes (68.18%) and 41 out of 68 apoptosis-related genes (60.29%) in group 2. In addition, Ino80 bound to the TSS of 17 out of 27 cell cycle-related genes (62.96%) and 41 out of 70 apoptosis-related genes (58.57%) in group 4 (Figure 2B). Genes (RB1, Cdkn1a, Cdkn2a, Mdm2, and Cdkn2b) enhanced by Ino80 knockout (KO) in group 4 were mainly negative cell cycle regulators of the G1-phase. Representative genes in group 2 (Ccne1 and Cdkn1b) and group 4 (Rb1 and Cdkn1a) are shown in Figure 2C. Although the difference in levels of gene expression by Ino80 deletion was subtle in naïve ESCs, the decrease (group 4) or increase (group 2) in gene expression by Ino80 deletion was much greater in primed ESCs. Gene ontology enrichment analysis (GO term analysis) for genes revealed that Ino80 directly binds to TSS in groups 2 and 4, indicating “G1/S transition of mitotic cell cycle”, “positive regulation of apoptotic process”, and “negative regulation of apoptotic process” (Figure 2D).

### 2.3. Abnormal Cell Cycle Profile Is Persistent in Differentiating ESCs

We sought to determine whether Ino80-mediated cell cycle control persists in differentiating ESCs. Thus, both CONT and Ino80 iKO ESCs were subjected to differentiation toward the mesodermal lineage by culturing the cells in differentiating media with 4-OHTM. Both cells proliferated and formed embryoid bodies (EBs). However, differentiating ESCs from iKO EBs (referred to as Diff–iKO ESCs) after 4-OHTM treatment from day 5 grew slower than those from CONT EBs (referred to as Diff–CONT ESCs) (Figure 3A). To determine whether the cell cycle is impaired and apoptosis is induced in Diff–iKO ESCs, flow cytometry was performed. Interestingly, Diff–iKO ESCs displayed a mildly shortened G1-phase, significantly extended S-phase, and an increase in apoptotic cells (Figure 3B,C, Supplementary S3). Panels of flow cytometric analysis of cell cycle were included in Appendix A.

### 2.4. Ino80 Loss Impairs Expression of Genes Involved in Differentiation of Mesodermal Lineages

We performed an RNA-seq analysis for differentiating ESCs to understand the underlying molecular mechanism by which Ino80 potentially regulates cell proliferation, survival, and differentiation. Our analysis identified 1809 DEGs, of which 807 genes were upregulated and 1002 genes were downregulated by Ino80 KO (Figure 4A, Appendix A). Our analysis revealed that the genes involved in mesoderm-derived organogenesis were aberrantly expressed. In particular, GO analysis showed that the genes regulating development of heart, blood vessel, cartilage, and kidney were repressed in Diff–iKO ESCs (Figure 4A). GO terms were included in Appendix A. Representative genes associated with mesodermal lineage development are shown in Figure 4B. Immunofluorescence analysis revealed that cardiac lineage (cTnT(+) and alpha-tropomyosin (+)) cells were reduced in Diff–iKO ESCs (Figure 4C).

### 2.5. Conserved and Unique Alteration of Gene Expression between Undifferentiated and Differentiating iKO ESCs

We examined whether genes differentially expressed by Ino80 KO in ESCs are also altered in Diff–iKO ESCs. As shown in Figure 5A, 22 and 138 out of 801 DEGs in group 2 were upregulated and downregulated by Ino80 depletion, respectively, whereas 96 and 14 out of 622 DEGs in group 4 were upregulated and downregulated by Ino80 depletion, respectively (Figure 5A). Interestingly, subsets of genes categorized into cell cycle- or apoptosis-related GO terms observed in serum–iKO ESCs were also altered. Furthermore, our analysis showed that genes related to mesoderm lineage differentiation were significantly altered in Diff–iKO ESCs (Figure 5B, Appendix A). Therefore, our analysis established that Ino80 function is required for the regulation of not only lineage-controlling genes, but also cell cycle-controlling genes in both serum–iKO ESCs and Diff–iKO ESCs. However, our analysis uncovered that, unlike serum–iKO ESCs, Ccne1 and Anxa11 included in group 2 were upregulated in Diff–iKO ESCs (Figure 2B and Figure 5C). These genes affect the G1/S transition, and this may explain the different phases of cell cycle arrest in serum–iKO ESCs and Diff–iKO [30,31,32].

## 3. Discussion

Ino80 is a member of the ATP-dependent chromatin remodeler family that plays critical roles in diverse developmental processes [9,11,12,13,14,17,18,21,24,27,28]. Previous studies have demonstrated that Ino80 loss causes embryonic lethality due to severe growth retardation, and its function is required for transcriptional regulation, DNA replication, DNA damage repair, and genome stability [15,23]. Moreover, several genetic studies have shown that Ino80 loss leads to impaired cell proliferation and cell death [11,13,14,17,18,22,23,26,27,28]. Previously, Ino80 function was shown to be required for the regulation of cell cycle in yeast [33,34].

Ino80 contributes to the regulation of pluripotency- or differentiation-related transcription by binding to TSS [14,24,28]. For example, a recent study on mESCs revealed that cells lacking Ino80 had a lower self-renewal ability with decreased expression levels of pluripotency markers and more dynamic expression levels of lineage markers [24,28]. Its function in ESC maintenance seems to be clearly differential depending on culture conditions, as variability in gene expression change was more prominent in serum–ESCs than in 2i–ESCs following Ino80 depletion [14,28]. Yu et al. showed that this tendency is tightly controlled by an epigenetic mark of H3K4me3 and H3K27me3 bivalency [28]. In addition to the Ino80 function in transcriptional control, it was shown that the more ESC transitioned from the naïve state to the primed state, the greater the cell loss caused by Ino80 depletion [14,28]. These findings imply that Ino80 is required for primed ESCs to maintain pluripotency, suppress differentiation, and properly proliferate. However, the mechanism by which Ino80 regulates ESC maintenance remains unclear. Furthermore, its regulatory role in differentiating ESCs has not yet been defined.

In the present study, the retention in the G1-phase of 2i–iKO ESCs was considerably longer than that of serum–iKO ESCs, which displayed an extended cell cycle compared with 2i–ESCs (Figure 1G). We showed that Ino80 loss extended the period of G1-phase more in serum–ESCs by upregulating RB1, Cdkn1a, Cdkn2a, and Cdkn2b, which are known to induce G1 arrest (Figure 2B). It has previously been demonstrated that Cdkn1a is negatively regulated by Ino80 [10]. Interestingly, the majority of the cell cycle and apoptosis genes belong to genes with bivalent domains (data not shown) [28]. However, Ino80 also promoted the expression of certain cell cycle regulatory genes (Ccne1, Cdc25b, and Cdkn1b). These findings are consistent with previous studies that showed that Ino80 contributes to either transcriptional activation or repression in a context-dependent manner [17,24,27,28].

RB function is critical, but not necessary, for the regulation of ESC cycle. A lower RB activity in 2i–ESCs with a longer G1-phase indicates that alteration of RB activity by Cdkn1a and Cdkn1b is critical to extend the G1-phase in ESCs [4]. In contrast, Wirt et al. demonstrated that G1 arrest in mESCs can occur via an RB-independent pathway [35]. Aberrant RB function leads to an abnormal cell cycle in various cancer cells. It has been shown that mutations in both RB and CDKN2A leads to resistance of rapamycin-mediated cell death in breast cancer cells. RB, along with p14^ARF^, the CDKN2A gene product, suppresses E2F family transcription factors that promote cell cycle progression from G1 to S phase in breast cancer cells [36]. Unlike in serum–ESCs, enhanced Cdkn1a expression by Ino80 knockdown in 293T cells causes G2 arrest; thus, Cdkn1a potentially regulates not only G1/S transitions but also G2/M transitions [10]. G1 arrest was more prominent in pRb-positive cells, which is in accordance with the prolonged G1 arrest observed in serum−ESCs [4,20]. An elevated level of Cdkn1b, like that of Cdkn1a, causes G1 arrest when Cdkn1b is upregulated in HeLa S3 cells [37]. However, our analysis showed a decreased expression of Cdkn1b in iKO serum–ESCs. This may be explained by the fact that the expression of Ccne1, a main target of Cdkn1b, is reduced by Ino80 KO, which in turn induces G1 arrest [38]. Thus, Ccne1 is one of the targets of E2F, which is regulated by RB [39].

Therefore, the present study, along with other previous studies, suggests that Ino80 has an Rb-dependent function in the cell cycle of ESCs by changing G1-phase regulators, including Cdkn2a, Cdkn1a, Ccne1, and Cdkn1b [4]. With regard to cell apoptosis, however, both “positive regulation of apoptotic process” and “negative regulation of apoptotic process” were simultaneously detected in serum–iKO ESCs (Figure 2D). The apoptotic phenotype in serum–ESCs may be caused by either Ino80 deletion-mediated G1 arrest or additional factors. Murada et al. showed that G1 arrest is associated with apoptosis in human breast cancer cells [40], and Keil et al. demonstrated that a longer S-phase results in a higher rate of apoptosis after Ino80 deletion in neural progenitors [27]. These results are consistent with the phenotypes of Ino80 KO embryos, which can undergo implantation but fail to develop beyond the egg cylinder stage, corresponding to primed ESCs [14].

Qiu et al. showed that Ino80-depleted ESCs nearly fail to differentiate, form teratomas in vivo, and exhibit incomplete formation of EBs in vitro. Our differentiation of iKO ESCs in normal differentiation media also failed to generate EBs of a homogeneous size [14]. Thus, in our differentiation experiment, we replaced the normal differentiation medium with medium containing KnockOut™ Serum Replacement (KSR), which can efficiently produce EBs [21].

Diff–iKO ESCs exhibited limited differentiation potential and low proliferation rate. The Ino80 KO effect is associated with not only aberrant transcriptional regulation but also genome instability in various differentiated cells [11,17,18,22,27]. Ino80 represses the expression of trophoblastic invasion-related genes during trophoblast development, and recurrent miscarriages can occur when Ino80 is suppressed [18]. During neural progenitor cell (NPC) differentiation, genes involved in double strand break (DSB) repair pathways are mainly upregulated by Ino80 deletion, but NPC differentiation-related genes are suppressed [27]. The function of Ino80 is to regulate gene expression in developing spermatocytes. Thus, Ino80 deficiency in murine spermatocytes causes failure of spermatogenesis; it leads to an aberrant repair of DNA double-strand breaks, resulting in apoptosis [17]. Moreover, Ino80 occupies E2F target gene promoters, suppresses their expression, and increases S-phase occupancy, and its loss in vascular endothelial cells causes defects in cardiac compaction [11]. In the present study, the DEGs (Hoxb9, Cav1, Calcrl, Fat4, and Mgp) in Diff–iKO ESCs were mainly included in the GO terms of mesodermal lineage development (Figure 4A). Ino80-silenced MSC grown in an osteogenic environment exhibit decreased mRNA levels of osteoblast-specific genes [12]. Interestingly, differentiation-related genes that were upregulated by Ino80 in primed ESCs were downregulated in Diff–iKO ESCs (Figure 5B), suggesting that regulatory mechanism(s) of cell cycle- and differentiation-related genes by Ino80 in primed ESCs appear similar, but they could be different in differentiating ESCs.

Our analysis showed that the differentiating ESCs also aberrantly expressed cell cycle genes in primed ESCs and exhibited increased apoptosis (Figure 3B and Figure 5B). Interestingly, Diff–iKO ESCs exhibited an extended S-phase compared with Diff–CONT ESCs (Figure 3C). The prolonged S-phase found in differentiating ESCs is in line with earlier research showing that the Ino80 complex activates the transcription of S-phase genes by binding to mlui-binding factor (MBF)-regulated promoters in yeast, and Ino80 accelerates the S-phase by decreasing the expression of E2F target gene in differentiated cells [11,33]. Interestingly, our RNA-seq analysis of Diff–iKO ESCs revealed a conserved Ino80 function in the regulation of cell cycle and apoptosis, as some of the group 2 and 4 genes were identified in Diff–iKO ESCs. Moreover, the expression pattern of Cdkn1a and Cdkn1b was maintained in Diff–iKO ESCs. However, we observed altered expression patterns of Ccne1 and Anxa11 in Diff–iKO ESCs, which differed from those in serum–iKO ESCs, and they may be associated with the longer S-phase (Figure 3C and Figure 5C). S-phase is prolonged when CCNE1 is overexpressed or amplified in an E2F dependent manner in ovarian cancer cells [30,31]. The downregulation of Anxa11 in AGS and SGC-7901 cell lines leads to a significant decrease in the percentage of cells in the S-phase [32].

However, although the apoptotic rate was increased in Diff–iKO ESCs, no corresponding DEGs were found, suggesting that apoptosis may have occurred via an unknown mechanism. Aberrant DNA repair by Ino80 loss may be one of these mechanisms [13]. Alternatively, Xu et al. proposed that apoptosis might be caused by S-phase arrest [41].

## 4. Materials and Methods

### 4.1. Animals and Generation of Ino80 iKO ESCs

All mouse strains were bred at Konkuk University, Korea (IACUC#; KU21020). Ino80 conditional allele mice (stock# 07816) were purchased from ICS, Institute Clinique de la Souris, and Gt (ROSA26^CreER^) (stock# 004847) mice were purchased from Jackson Laboratory (Bar Harbor, USA). The CF-1 mouse strain used for the production of feeder cells was purchased from the Laboratory Animal Resource Center in Korea Research Institute of Bioscience and Biotechnology (KRIBB). To produce Ino80^2f/2f^ ROSA26^CreER^ mouse embryos (referred to as Ino80 iKO), female Ino80^2f/2f^ mice were crossed with male Ino80^2f/+^ ROSA26^CreER^ mice. For timed mating, the vaginal plug was checked, and female mice were sacrificed at E3.5 to generate Ino80 iKO mESCs. At E3.5, blastocysts were harvested from the uterus in M2 medium (Sigma-Aldrich, St. Louis, MO, USA) and placed in pronase to remove the zona pellucida. Zona pellucida-free blastocysts were transferred to a culture dish containing mitomycin-C-treated mouse embryonic fibroblast (MEF). Ino80^2f/2f^ ROSA26^CreER^ blastocysts were used as the experimental group and Ino80^2f/2f^ blastocysts were used as the control group in this study. The established ESCs were maintained in serum-free 2i/LIF culture condition (2i) or serum [15% fetal bovine serum (FBS) (S001-01; Welgene, Gyeongsan-si, Gyeongsangbuk-do, Republic of Korea)] culture condition on a 0.2% gelatin-coated dish.

### 4.2. ESC Differentiation

For EBs formation, ESCs were induced to the primed state in serum-medium for two days. Then, 500,000 ESCs were suspended in 5 mL KSR medium [Dulbecco’s Modified Essential Medium (DMEM) (SH30243.01; Hyclone, Logan, UT, USA) supplemented with 8% KSR (10828028; Gibco, Carlsbad, CA, USA), penicillin/streptomycin (Gibco, Carlsbad, CA, USA), and β-mercaptoethanol (Gibco, Carlsbad, CA, USA)] to allow them to form EBs. The EBs were fed every 2 days. At day 4 of EBs formation, the EBs were transferred to a 0.2% gelatin-coated culture dish containing KSR medium and allowed to attach. The following day, EBs were fed with differentiation medium [DMEM (Hyclone, Logan, UT, USA) supplemented with 15% FBS (S001-01; Welgene, Gyeongsan-si, Gyeongsangbuk-do, Korea), Minimum Essential Medium (MEM) non-essential amino acids (Gibco, Carlsbad, CA, USA), penicillin/streptomycin (Gibco, Carlsbad, CA, USA), sodium pyruvate, β-mercaptoethanol (Gibco, Carlsbad, CA, USA), and BMP4 (314-BP-010; R&D Systems, Minneapolis, MN, USA)] with 0.5 µM 4-OHTM (H6278; Sigma-Aldrich, St. Louis, MO, USA), and cultured for 7 days.

### 4.3. PCR Genotyping

For genotyping, mESCs or mouse tissues were lysed in 25 mM NaOH at 95 °C for 30 min or 2 h, respectively. The lysates were neutralized with 1 M Tris-Cl. PCR was performed using e-Taq DNA polymerase (SET15-R500; Solgent, Yuseong-gu, Daejeon, Republic of Korea) with the following conditions: pre-denaturation for 5 min at 94 °C, 35 cycles of denaturation for 30 s at 94 °C, primer annealing for 30 s at 58 °C or 62 °C, extension for 30 s at 72 °C, and a final extension for 7 min at 72 °C. Electrophoresis was performed on a 2% agarose gel, and PCR products were confirmed using UV light. The following primer sets were used for the detection of null allele and ROSA-CreER gene: Ino80 null primer (forward 5′-GGC AGC CCA AGG TAG ACT CAG CC-3′, reverse 5′-AGG CCT TAT TTA GCT CAG GTT GGC −3′) (amplicon size, WT allele: 879 bp and KO allele: 248 bp) and ROSA-CreER primer (forward 5′-CAT GAA CTA TAT CCG TAA CCT GGA-3′, reverse 5′-CAT CCA ACA AGG CAC TGA CCA TCT-3′) (amplicon size, 197 bp).

### 4.4. Analyses of Cell Proliferation, Cell Cycle, and Apoptosis

After seeding 20,000 4-OHTM-treated cells on 0.2% gelatin-coated plates, the cells were harvested and counted on days 2, 3, and 4. Cell counting was performed using the automatic cell counter EVE^TM^ (EVE-MC; NanoEntek). For cell cycle analysis, the cells were fixed with 80% ethanol at 4 °C for ~16 h and treated with RNase A solution (R6148; Sigma-Aldrich, St. Louis, MO, USA) for 2 h after washing with ice cold Dulbecco’s Phosphate-Buffered Saline (DPBS). Then, the cells were stained with propidium iodide (556463; BD Pharmingen, Franklin Lakes, NJ, USA) and subjected to flow cytometric analysis using CytoFLEX (Beckman Coulter, Brea, CA, USA). For the measurement of apoptotic cells, an FITC-conjugated Annexin V Apoptosis Detection Kit I (556547, BD Pharmingen, Franklin Lakes, NJ, USA) was used.

### 4.5. Immunofluorescence

The cells were fixed in 4% Paraformaldehyde (PFA) in Phosphate-buffered saline (PBS) at room temperature (RT) for 5 min. The fixed cells were permeabilized in PBS with 0.2% Triton X-100 at RT for 5 min. Then, the cells were incubated for 1 h at RT in blocking solution (3% donkey serum and 2% bovine serum albumin in PBS). Cells were incubated overnight with primary antibodies in blocking solution at 4 °C. The primary antibodies used in the study were anti-cTnT (ab8295; Abcam, Cambridge, UK) and anti-alpha-trophomyosin (ab109505; Abcam, Cambridge, UK). The cells were washed thrice with 0.1% Tween20 in PBS and then incubated with secondary antibodies for 1 h at RT. To visualize the nuclei, 4′, 6-diamidino-2-phenylindole (DAPI) was diluted in PBS (1:1000). Images were captured using a confocal microscope Zeiss LSM800.

### 4.6. Quantitative RT-PCR Analysis

Total RNAs from cells were extracted using RNeasy Plus Mini Kit (74104; Qiagen, Hilden, Germany), and cDNA was synthesized using the TOPscript RT DryMix cDNA synthesis kit (RT211; Enzynomics, DE, KR) according to the manufacturer’s instructions. The following primer sets were used for qRT-PCR: mGAPDH primer (forward 5′-CAT GGC CTT CCG TGT TCC TA-3′, reverse 5′-GCC TGC TTC ACC ACC TTC TT-3), mIno80 primer (detecting exon 6) (forward 5′-AAG GAG TTG CAG CAG TAC CA-3′, reverse 5′-TCT TCT TTT TGG GTC CAA GC-3′), and mCdkn1a primer (forward 5′-TTG CAC TCT GGT GTC TGA G-3′, reverse 5′-GTG ATA GAA ATC TGT CAG GCT G-3′). qRT-PCR was performed in StepOnePlus™ Systems (Applied Biosystems, Waltham, MA, USA) using Fast SYBR^®^ Green Master Mix (4385616; Applied Biosystems, Waltham, MA, USA). The expression levels of the target genes were normalized with GAPDH level in each sample.

### 4.7. RNA-seq and ChIP-seq Analysis

Total RNA samples were prepared from differentiating ESCs (Diff–CONT and Diff–iKO ESCs) using RNeasy Kits (74,104; Qiagen, Hilden, Germany). For RNA-seq library preparation, 1 µg of total RNA was used for the preparation of the RNA-seq library using the TruSeq Stranded mRNA Sample Preparation Kit. Paired-end sequencing reads were produced on illumina NextSeq 500 sequencing platform.

RNA-seq (GSE158545) and ChIP-seq (GSE158545) were downloaded from the NCBI Gene Expression Omnibus (GEO) [28] and mapped to the mm10 mouse genome assembly using STAR (v2.7.9a) with the default option [42]. FPKM values were generated using the Cuffquant and Cuffnorm tools of the Cufflinks toolset (v2.2.1) [43]. DEGs were determined as FC ≥ 2 and FPKM ≥ 2. The heatmap was generated using the heatmap.2 tool in R [44] with FPKM values of selected DEG and clustered with the hclust tool in R [43]. GO term analysis was conducted using the DAVID web tool (version 2021) [45]. The GO term results were visualized using the GO circle package or ggplot2 (bubble plot) in R. Raw ChIP-seq reads were mapped to the mm10 mouse genome assembly using Bowtie2 (2.3.5.1) [46]. The mapping file (SAM file) was converted into a BAM file and sorted using the same tools (1.11) [47]. Peaks were detected using the MACS2 tool [48] and then applied to MACS2 bdgcmp to remove noise signals. Peak annotation to the “promoter-TSS” for each sample was obtained using Homer annotation peaks [49]. RNA-seq and ChIP-seq data were visualized using the Integrative Genomics Viewer (IGV) tool [50].

### 4.8. Statistical Analysis

All statistical analyses were performed using GraphPad Prism version 8.0.2 (GraphPad Software Inc., La Jolla, CA, USA). One-way ANOVA with Tukey’s test as post hoc analysis was used for Ino80 qRT-PCR data, apoptosis, and cell cycle analyses in ESCs. Unpaired *t*-test analysis was performed on the remaining data. The error bars in the data represent the standard error of the mean (SEM). Statistical significance was determined at a *p*-value lower than 0.05 (*), 0.01 (**), 0.001 (***) or 0.0001 (****).

## Figures and Tables

**Figure 1 ijms-23-15402-f001:**
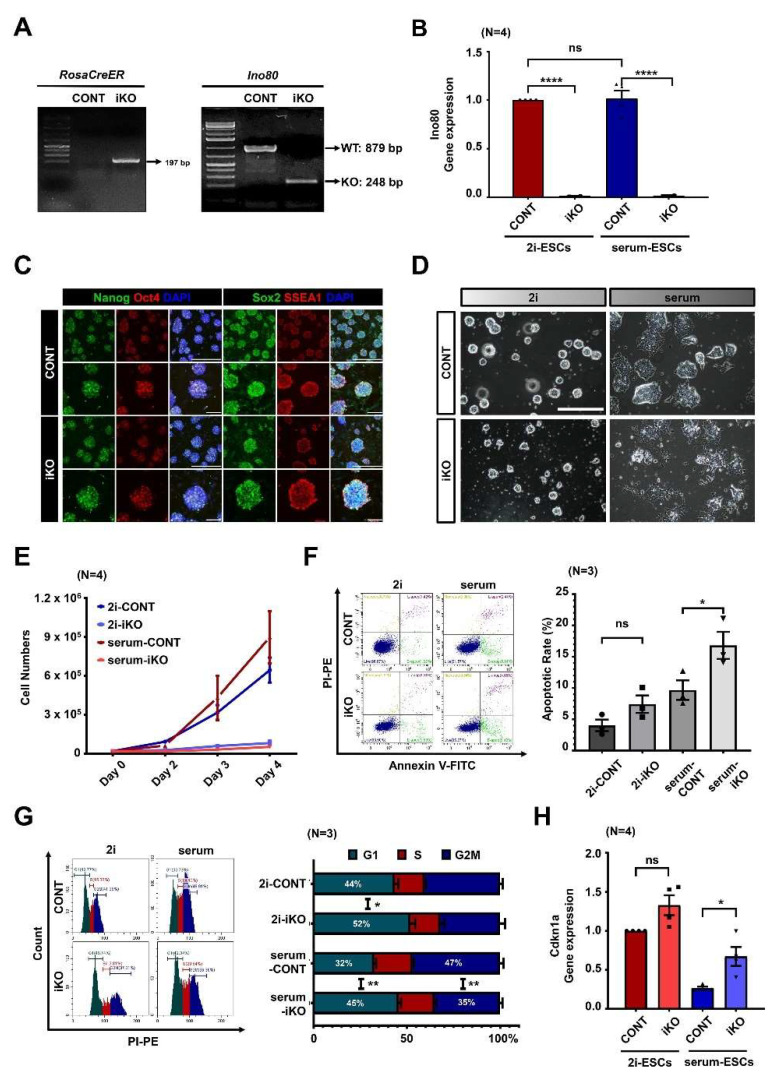
Ino80 loss leads to extension of G1-phase and apoptosis in embryonic stem cells cultured in serum-media (serum–ESCs). (**A**) Result of PCR genotyping. ROSA26-CreER allele was detected only in Ino80 iKO ESCs. After treating the ESCs with 4-OHTM, homozygous conditional allele was detected in CONT ESCs and Ino80 knockout allele was detected in Ino80 iKO ESCs. Ino80 iKO ESCs, inducible Ino80 knockout ESCs. 4-OHTM, 4-hydroxytamoxifen. ESCs, embryonic stem cells. CONT ESCs, control ESCs. (**B**) Expression levels of Ino80 using mIno80 primer (detecting exon 6) in both CONT and Ino80 iKO ESCs maintained in 2i- or serum-media was examined by qRT-PCR (N = 4). mIno80, mouse Ino80. (**C**) Immunofluorescence analysis of pluripotent markers (Nanog, Sox2, and SSEA1) in CONT and Ino80 iKO ESCs cultured in 2i-media containing 4-OHTM. Scale bar, 200 μm. (**D**) Representative images of CONT and Ino80 iKO ESCs cultured in either the 2i- or serum-media. ESCs were treated with 4-OHTM for 3 days and adapted to 2i- and serum-media for at least 2 days. Scale bar, 200 μm. (**E**) Cell proliferation assay. Cell numbers of the CONT and Ino80 iKO ESCs were counted at days 2, 3, and 4 after seeding 20,000 ESCs cultured in either 2i- or serum-media (N = 4). (**F**) Representative images of apoptosis analysis by flow cytometry and quantification. Annexin V(+) cells were measured in the CONT and Ino80 iKO ESCs. The difference in apoptosis rates between CONT and Ino80 iKO ESCs cultured in 2i-media is 3.40% on average, and in serum-media is 7.14% on average. *p*-value was calculated by one-way ANOVA test (N = 3). (**G**) Representative histograms of cell cycle and quantification. Percentages of G1, S, and G2/M phases of the CONT and Ino80 iKO ESCs measured by flow cytometry were plotted (N = 3). (**H**) Relative expression of Cdkn1a, by using qRT-PCR in both 2i-media and serum-media, the expression levels of Cdkn1a in CONT and Ino80 iKO ESCs were investigated (N = 4). Statistical significance was determined at a *p*-value lower than 0.05 (*), 0.01 (**) or 0.0001 (****).

**Figure 2 ijms-23-15402-f002:**
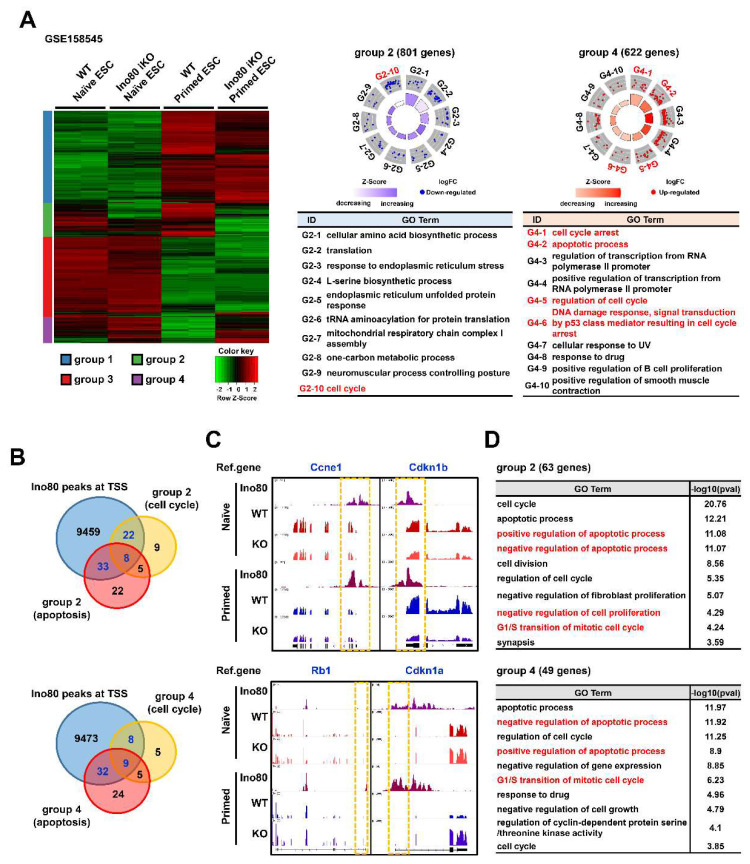
Ino80 directly regulates genes involved in cell cycle. (**A**) Heatmap and GO circle plot showing DEGs and GO terms of GSE158545. Unsupervised grouping with 5483 DEGs in total (cutoff value, FC ≥ 2, FPKM ≥ 2) resulted in 4 groups; group 1 (2170 genes), group 2 (801 genes), group 3 (1890 genes), and group 4 (622 genes). GOBP for group 2 and 4 genes were visualized through GO circle tool. Cell cycle- and apoptosis-related terms are highlighted in red fonts. DEGs, differentially expressed genes. GO terms, gene ontology terms. FC, fold change. FPKM, fragments per kilobase of transcript per million. GOBP, Gene Ontology Biological Process. (**B**) Venn diagram showing common genes among cell cycle, apoptosis, and Ino80 targets in the group 2 and 4. “Ino80 peaks at TSS” gene was defined as a gene which Ino80 was detected in TSS (GSE158545). Common genes are highlighted in blue fonts. TSS, transcription start site. (**C**) Visualization of representative cell cycle-related genes with Ino80 bindings and expression levels using the IGV. Yellow dotted line indicates TSS. IGV, Integrative Genomics Viewer. (**D**) GO term analysis with genes that Ino80 binds to TSS and are involved in cell cycle or apoptosis in group 2 (63 genes) and group 4 (49 genes). The tables show the GO terms and −log10(*p*-value). Cell cycle- and apoptosis-related terms are highlighted in red fonts.

**Figure 3 ijms-23-15402-f003:**
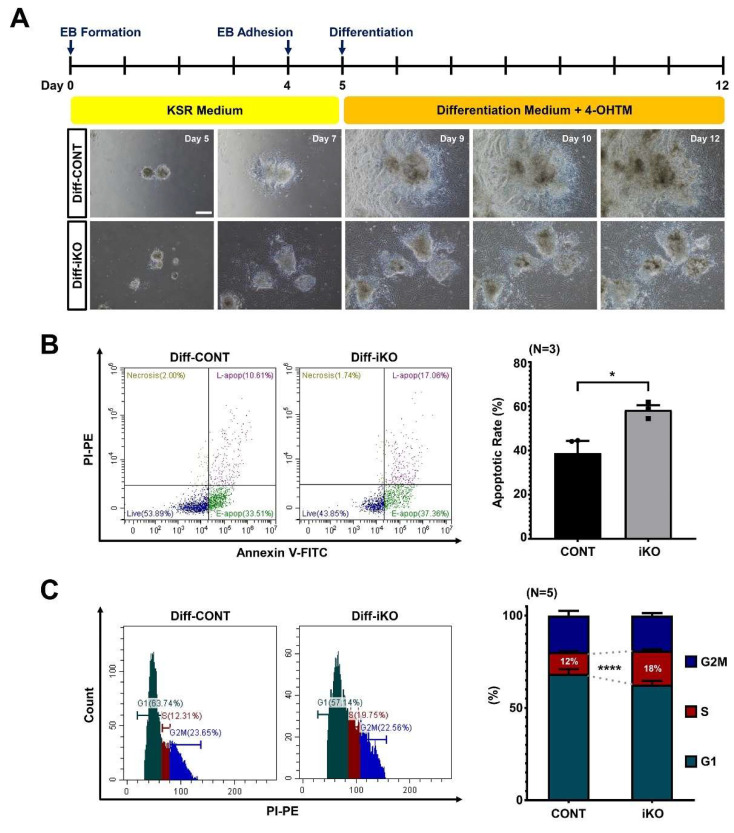
Ino80 depletion in differentiating ESCs causes extension of S-phase and apoptosis. (**A**) Schematic images showing differentiation protocol of differentiating ESCs, and representative images showing expansion of differentiating cells from for EBs. Images were captured at day 5, 7, 9, 10, and 12 after seeding same numbers of EBs on gelatin-coated dishes. Scale bar, 200 μm. EBs, embryoid bodies. (**B**) Representative images of apoptosis analysis in Diff−CONT and Diff−iKO ESCs and quantification (N = 3). The difference in apoptosis rates between Diff−CONT and Diff−iKO ESCs is 19.57% on average. Diff−CONT ESCs, differentiating ESCs from CONT ESCs. Diff−iKO ESCs, differentiating ESCs from Ino80 iKO ESCs. (**C**) Representative cell cycle histogram of Diff−CONT and Diff−iKO ESCs and their percentages of G1, S, and G2/M phases (N = 5). Statistical significance was determined at a p-value lower than 0.05 (*) or 0.0001 (****).

**Figure 4 ijms-23-15402-f004:**
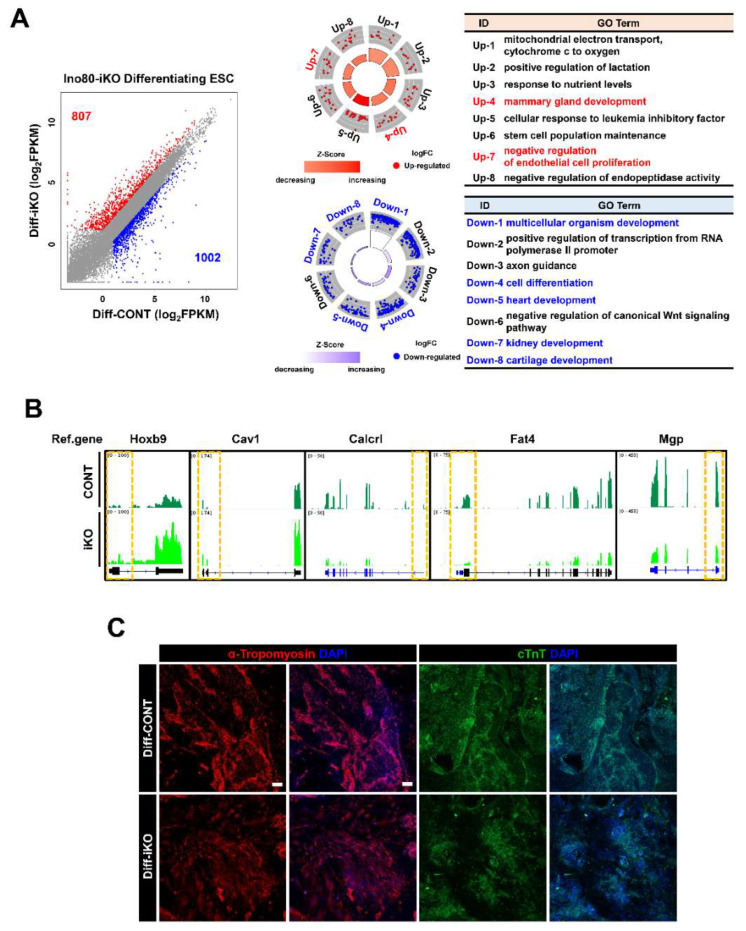
RNA-seq analysis revealed aberrant expression of genes important for organ development in Diff–iKO ESCs. (**A**) Scatter plot showing DEGs in Diff–iKO ESCs, and GO term analysis. Total 1809 DEGs (Cut-off value, FC ≥ 2, FPKM ≥ 2); 807 up-regulated genes and 1002 down-regulated gene were detected. GO terms associated with differentiation are highlighted with red or blue fonts. (**B**) Visualization of representative DEGs (Hoxb9, Cav1, Calcrl, Fat4, and Mgp) related to mesodermal differentiation GO terms in Diff–iKO ESCs using IGV. Yellow dotted line indicates TSS of the corresponding genes. (**C**) Immunofluorescence of cardiac muscle marker (cTnT and alpha-Tropomyosin) in Diff–CONT and Diff–iKO ESCs. Scale bar, 200 μm.

**Figure 5 ijms-23-15402-f005:**
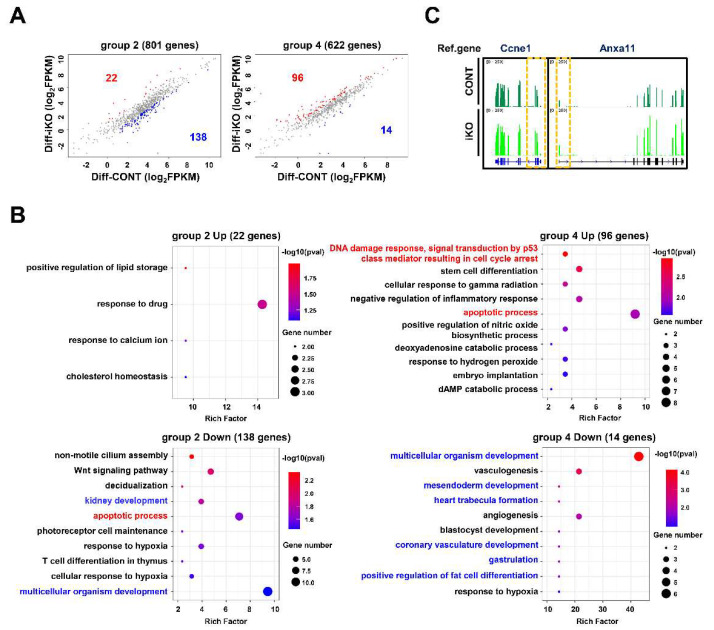
Cell cycle- and apoptosis-related genes in differentiating ESCs. (**A**) Scatter plots showing selected DEGs (that are contained in group 2 and 4 in Figure 2A) in the Diff–iKO ESCs. (**B**) Bubble plots showing GO terms with group 2 or 4 DEGs that are common up-regulated or down-regulated genes in Diff–iKO ESCs. Terms associated with cell cycle or apoptosis are highlighted with red fonts, and terms associated with development are highlighted with blue fonts. (**C**) Visualization of representative DEGs (Ccne1 and Anxa11) associated with G1/S transition using IGV.

## Data Availability

GSE214162 for RNA-seq data.

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
