# Peer review of "INO80 Is Required for the Cell Cycle Control, Survival, and Differentiation of Mouse ESCs by Transcriptional Regulation"

_ijms, 2022, doi:10.3390/ijms232315402_

Round 1

Reviewer 1 Report

In this study, the authors reported the loss of Ino80, a chromatin modifier, impairs cell cycle progression in proliferating and differentiating ESCs. I appreciate the work of generating conditional knockout ESCs, which allows temporal KO of Ino80 in specific days of ESC culture. However, the novelty, originality, and data quality are subpar. The experiments fail to resolve the exact role of Ino80 in regulating cell cycle progression and ESC differentiation. Mechanistic insight is absent. The design of appropriate controls are crucial but here, the authors completely skip the control for CreER and tamoxifen/4-OHT treatment. This is particularly essential as numerous studies have pointed out CreER alone could hamper stem cell proliferation. Importantly, data quality and presentation are poor, making it difficult for audience to interpret the results. Given the amount of extra work that will be needed to improve the manuscript to a standard that warrant publication in IJMS, I recommend the manuscript to be rejected. Please see detailed comments below.

1. The author claimed, this study “established a mechanistic link between the function of chromatin remodeler and control of ESC cell cycle”. This is an ambitious overstatement. First, the authors did not perform experiments that can directly demonstrate the chromatin structure in Ino80KO ESCs, e.g. HiC, ATAC-seq. Instead, the authors performed RNA-seq. One must know that any transcriptional changes may only reflect “what is happening” or “what happened” in the cell, but not “what causes it to happen”. The ChIP seq data is downloaded from publicly accessible database and not originally generated in this study. Simply put, this study only provides associative clues, not mechanistic insights. Please consider tone down all statements concerning “mechanistic”, “chromatin remodeling”, etc. in the manuscript.

2. Throughout the study, Ino80f/f ESCs are used as controls, but this is not sufficient. A control of Ino80ff;R26R-CreER without tamoxifen/4-OHT treatment should be included, at least in key experiments, to rule out the possibility that any difference observed is due to CreER instead of Ino80 KO.

It is well-known that activation of CreER itself alone, without knocking out any genes, impairs stem cell proliferation (PMID 30449703 Bohin et al.)

3. Please include all individual data points in the bar charts to allow fair statistical evaluation. For example, in fig. 1F and 1H, the different looks substantial but it’s “ns”. Individual data points with the bar will help audience to interpret the results.

4. Figure 3B, the authors reported ~40% cells are apoptotic in the control. This is way higher than in a normal case.        Are there any reasons for this? Or simply this is an experimental artifact that the cells are cultured in an unhealthy state.

5. In Figure 3C, it seems that a significantly larger population of iKO cells are loaded to the flow cytometer compared to controls. This makes it hard to interpret the graph and is for sure an unfair comparison.

6. In Fig.4, given that the authors reported in Fig. 3 that there is remarkable difference in cell cycle states and also apoptotic rate, could the DEG from RNA-seq only reflecting such changes in cell cycle behaviour? Or, these DEGs are only the consequences of shortened G1 and lengthened S phase?  The authors should FACS the cells based on cell cycle states and perform the RNAseq, this will give a clearer resolution to the RNAseq result but not just mere inference from GO term analysis.

7. Also, Figure legends of 4B and 4C are swapped. Immunofluorescence is in Fig. 4C not 4B.

8. The immunofluorescence images look suspicious. While the authors suggest cTnT and tropomyosin are down-regulated in the ino80 KO, the entire images including the DAPI look much dimmer in the KO panel than in the control. This implies the immunostaining or imaging condition is not consistent across different samples. Also, why is there grey signal in the cTnT+DAPI image?

9. I suggest not to use the term “cluster” for bulk RNAseq. This is misleading as one would assume “cluster” is now reserved for single cell rna seq in this era. This is especially misleading in fig. 5.

Author Response

Dear Reviewer 1,

The authors thank the reviewer so much for his/her valuable comments on the manuscript. Attached please see our point-by-point responses to your comments. Thank you for your time and consideration.

Best,

Kwonho 

Reviewer 2 Report

The paper examines the significance of INO80 as an ATP-dependent chromatin remodeler that plays indispensable roles in development, cancer progression, and gene transcription. The authors confirmed that Ino80 loss in ESCs causes an aberrant cell cycle and cell death. The methodology is well explained, and I believe that the paper can be accepted for publication in the journal.

Author Response

Dear Reviewer 2,

The authors thank the reviewer so much for his/her support on our study.

Best,

Kwonho

Round 2

Reviewer 1 Report

The current manuscript is significantly improved. The authors have addressed most of my concerns. However, I still believe the key experiments in the study, for example, the cell cycle analysis by flow cytometry, would need 2 more controls to make it legit: (1) Ino80ff;R26R-CreER without 4-OHTM and (2) Ino80 ff with 4-OHTM treatment. We need to bear in mind that it is still controversial whether tamoxifen/4-OHTM or CreER would alter cell cycle properties. These are easy and straightforward experiments that can be finished in a pretty short timeframe (<2 weeks). Even for in vivo studies,  tamoxifen control or CreER control is required as a standard. Given that the cell cycle property alteration is the major finding in this study, it is rational to rule out any possibilities that is due to potential experimental artefacts. 

Author Response

Dear reviewer,

Attached please see our response to your comment. The authors thank you so much for the insightful comment and your support.

Best wishes,

Kwonho

Round 3

Reviewer 1 Report

I recommend acceptance for publication. 

Author Response

Thank you so much.